# Development of Microalgae Biosensor Chip by Incorporating Microarray Oxygen Sensor for Pesticides Sensing

**DOI:** 10.3390/bios9040133

**Published:** 2019-11-12

**Authors:** Md. Abul Kashem, Kazuki Kimoto, Yasunori Iribe, Masayasu Suzuki

**Affiliations:** Department of Electric and Electronic Engineering, Faculty of Engineering, University of Toyama, Toyama 930-8555, Japan; sobujus@gmail.com (M.A.K.); suzukimy18454@gmail.com (K.K.); iribe@eng.u-toyama.ac.jp (Y.I.)

**Keywords:** microalgae, pesticides, biosensor chip, fluorescence oxygen sensor

## Abstract

A microalgae (*Pseudokirchneriella subcapitata*) biosensor chip for pesticide sensing has been developed by attaching the immobilized microalgae biofilm pon the microarray dye spots (size 100 μm and pitch 200 μm). The dye spots (ruthenium complex) were printed upon SO_3_-modified glass slides using a polydimethylsiloxane (PDMS) stamp and a microcontact printer (μCP). Emitted fluorescence intensity (FI) variance due to photosynthetic activity (O_2_ production) of microalgae was monitored by an inverted fluorescent microscope and inhibition of the oxygen generation rate was calculated based on the FI responses both before and after injection of pesticide sample. The calibration curves, as the inhibition of oxygen generation rate (%) due to photosynthetic activity inhibition by the pesticides, depicted that among the 6 tested pesticides, the biosensor showed good sensitivity for 4 pesticides (diuron, simetryn, simazine, and atrazine) but was insensitive for mefenacet and pendimethalin. The detection limits were 1 ppb for diuron and 10 ppb for simetryn, simazine, and atrazine. The simple and low-cost nature of sensing of the developed biosensor sensor chip has apparently created opportunities for regular water quality monitoring, where pesticides are an important concern.

## 1. Introduction

The use of pesticides (including herbicides, insecticides, and fungicides) has increased dramatically in developing countries. For example, in Bangladesh, only 758 metric tons were used in 1960 but this increased to 48,690 metric tons in 2008 to allow the expected yield of crops to be achieved through the control of pests [1,2,3]. Pesticides in farming practices have offered many termed benefits, like frequent or year-round cropping, healthy (free from infection) grains and, also, expected yields, though at the expense of huge water contamination [4].

Although there are still debates regarding the actual number of pesticides that have been used all over the world, diuron, simazine, atrazine, simetryn, mefenacet, and pendimethalin have commonly been used by cultivators [5,6,7]. The threats of these pesticides to the environment as well as on organisms, including human beings, are important concerns and have become a major issue during the last decades [8,9,10]. The World Health Organization (WHO) and United Nations Environment Program (UNEP) have depicted approximately 4 million people suffering from severe pesticide poisoning and nearly 20,000 people are dying every year due to pesticide exposure, with the majority consistently in developing countries [11]. Several serious diseases or health effects in humans, e.g., cancer (thyroid and bone), reproductive and endocrine disruption, neurological damage, immune system dysfunction, etc., are now frequently occurring and a direct link with pesticide poisoning is suspected [12]. Thus, national and international regulations are encouraging minimizing or banning the use of pesticides. For an example, the European Union (EU), through their Water Framework Directive (WFD, 2000/60/EC) or the Urban Waste Water Directive (91/271/EEC), has strengthened water quality control and set a drinking water threshold of 0.1 μg/L for each individual pesticide and 0.5 μg /L for the sum of all pesticides [13,14].

In developed countries, environmental water quality monitoring is usually performed on a regular basis, but is often infrequent in many developing countries because of complications with the present laboratory methods, e.g., high-performance liquid chromatography (HPLC), gas chromatography coupled with mass spectrometry (GC/MS), which require highly skilled personnel, long times, and also high costs and, in addition, clash with the recent demands, such as for a simple and frequently usable method that is noninvasive, low cost, disposable, as well as can be integrated with other systems [5,13,15,16].

There are many efficient biosensors for toxicity sensing using different aquatic organisms like crustaceans, daphnia, algae, and fish (individually or as a battery of organisms). Microalgae (*Pseudokirchneriella subcapitata*) was identified as the most sensitive to any waterborne chemicals compared to others as well as being easy to culture and handle under laboratory conditions [17,18,19,20,21].

Through the photosynthetic activity of photosystem II (abbreviated as PSII), green microalgae convert light into chemical energy and generate oxygen as a byproduct. The photosynthetic activity is strongly inhibited if inhibitors (like pesticides) are present in their growth environment. Basically, pesticides are responsible for blocking the electron transport function in PSII and, thus, oxygen generation is inhibited. The inhibition of oxygen generation rate (%) and concentration of an individual pesticide often show good correlation [22,23].

Presently, the use of optical sensors is a better sensing technique for the evaluation of oxygen compared to electrochemical sensors, which are prone to being easily contaminated and integration is problematic. Optical sensors may also be fabricated in any shape and size, for example, in microarrays with fast and sensitive responses and this indeed saves costs and expands the possibility for application in many fields, and even the monitoring of biological functions at a single cell level [24,25,26,27,28,29].

Immobilization of microalgae is another important consideration that depends on the proper selection of immobilizing techniques and, hence, porous filter membranes like mixed cellulose ester (MCE), Omnipore®, etc., and their trapping techniques are always keep more advantages than other techniques [30].

In this study, we aimed to develop an optical pesticide-sensing microalgae biosensor chip based on the fluorescence intensity (FI) monitoring of ruthenium complex dye and FI, which always varies with the generation of oxygen according to the photosynthetic reaction of microalgae. Usually, the rate of FI decrease is high when the generated oxygen concentration is high, due to high quenching activity between fluorophores and oxygen molecules. In a nontoxic environment, microalgae continue to have photosynthetic activity under proper light supply conditions and generate oxygen, which reduces the FI, but within a toxic environment, the oxygen generation rate is inhibited and, hence, the decrease in FI is also limited. The decrease in FI before and after pesticide injection will be quantified as the inhibition of oxygen generation rate (%) against pesticides concentrations.

Pesticides biosensors using immobilized microalgae have been reported. The detection limit for diuron, for example, of these biosensors was from 2.5 to 100 ppb [31,32,33,34]. However, for use in developing countries, they are still expensive and complicated. In our present work, we have succeeded in sensing pesticides using our developed microalgae biosensor chip, which includes many advantages like having low cost, being disposable, providing fast responses, etc., and this sensor could be a potential sensing tool for water quality monitoring in developing countries.

## 2. Materials and Methods

### 2.1. Chemicals

Ruthenium complex dye, dichloro(tris-1,10-phenanthroline)ruthenium(II) hydrate, and Nafion® perfluorinated resin solution were obtained from Sigma-Aldrich (St.Louis, MO, USA). Simazine and atrazine were obtained from Wako Pure Chemical Industries Ltd (Osaka, Japan). Simetryn and pendimethalin were from G. L. Science (Tokyo, Japan), and mefenacet was from Kanto Chemical Co., Inc (Tokyo, Japan). All chemicals in this study were of analytical grade and used without further purification.

### 2.2. Microarray Oxygen Sensor Chip Development

#### 2.2.1. Stamp Mold Development

A half glass slide (size 38 × 26 mm^2^, Matsunami Glass Ind. Ltd., Kishiwada, Japan) was rinsed by liquid cleanser and acetone solution by applying ultrasonic beam followed by complete drying. After that, OmniCoat (Micro Chem) (a few drops) was used to coat the glass slide at 2500 rpm for 15 s by a spin coater (K-3591S-1, Kyowariken, Tokyo, Japan) and baked at 100 °C for 10 min within an oven (drying oven DO-300, Iuchi, Osaka, Japan). Then, SU-8 50 photoresist (Micro Chem) (few drops) was levelled upon the OmniCoat layer at 2500 rpm for 15 s and again baked at 100 °C for 30 min. After that, using UV mask aligner (Kyowariken, Tokyo, Japan), micropatterns of a photomask (spot size 100 μm, pitch 200 μm) (MESH, Osaka, Japan) were developed onto the SU-8 50 levelled glass slide (exposure time 60 s) and completed by post-baking at 100 °C for 10 min. The properly baked glass slide was then taken out from the oven and left for few minutes at room temperature (25 °C) for adequate cooling. Finally, the patterned glass slide was immersed into SU-8 developer solution (Micro Chem) (10 min) and gently shaken using a mild mixer (PR-12, Taitec, Koshigawa, Japan). Figure 1A shows the developed mold and its SEM image.

#### 2.2.2. Polydimethylsiloxane (PDMS) Stamp Development

The previously prepared mold was used as the master for developing the PDMS stamp by the systematic procedure described below. At first, Sylgard 184 silicon elastomeric base (Dow Corning) and Sylgard 184 silicon elastomeric curing agent (Dow Corning) were vigorously mixed (10:1) in a small plastic container until the entire mixture was filled with small bubbles, and this was then degassed by keeping it in an aspirator for 30 min. Furthermore, an orderly setting of different materials was fitted between the two metal stands. The whole process can be performed on a metal stand (lower), with a glass slide (bottom and up surfaces covered by 0.5 mm thickness silicone rubber (SR) sheets, AS ONE, Osaka, Japan) placed upon it, and a stamp mold was also tightly attached (patterned side up). Then, a 0.2 mm thick SR and 0.5 mm thick PET (polyethylene terephthalate) sheet with the middle position cutting space (larger size of the mold patterned area) were also attached on top. After that, the degassed silicone gel was placed upon the patterned area of the stamp mold (amount as required) and was again degassed by keeping it in a desiccator for 15 min. Subsequently, a PET sheet, glass slide (bottom and top surface covered by 0.5 mm thick SR sheet), and upper metal stand were attached orderly onto the degassed gel and, finally, clamped tightly. This completed arrangement was placed into an oven at 70 °C temperature for at least 2 h to harden the silicone gel. Upon completion of baking, it was left for cooling at room temperature and the developed PDMS stamp was then detached from the arrangement by soaking into the ethanol (100%) solution. Lastly, the image of the prepared PDMS stamp was taken by electron microscope, which is shown in Figure 1B.

#### 2.2.3. Ruthenium Complex Dye Solution Preparation

Oxygen sensing indicator dye solution was prepared by using ruthenium complex dye solubilized within PVA substrate-mixed MOPS buffer (5 mM, pH 7.0) and Nafion (5%) solution. The detailed processes are explained in our previous study [27].

#### 2.2.4. Microcontact Printing of Dye Solution

A PDMS stamp (size around 6 × 8 mm^2^) was treated by O_2_ plasma for 40 s in a quick autocoater (SC-701AT, Sanyu denshi, Tokyo, Japan) for creating a hydrophilic environment, and upon it, an approximately 10 μL drop of dye solution was place and dried with an air gun for 2 min or until the dye was almost dried out. Furthermore, with the help of a microcontact printer (μCP), the dyed stamp was stamped onto the SO_3_-modified glass slide. The printed dye spots were allowed to dry for several hours and checked by an inverted fluorescent microscope (Ex. 450–490 nm and Em. 520 nm and above, ECLIPSE, TE 2000-U, Nikon, Japan). These dye spots on the substrate are oxygen sensitive and, altogether, this ensemble is called the microarray oxygen sensor chip and is shown in Figure 1C.

### 2.3. Microalgae Culture and Suspension Preparation

Microalgae fresh water liquid culture media were prepared by mixing 75 mg Ca(NO_3_)_2_·4H_2_O, 50 mg KNO_3_, 25 mg disodium β-glycerophosphate pentahydrate, 20 mg MgSO_4_·7H_2_O, 0.05 μg vitamin B12, 0.05 μg Biotin, 5 μg thiamine HCl, 250 mg Tris(hydroxymethyl)aminomethane in 498.5 mL Direct-Q Millipore distilled water (DMDW) and 0.15 mL P-IV metal solution (100 mg EDTA.2Na, 19.6 mg FeCl_3_·6H_2_O, 3.6 mg MnCl_2_·4H_2_O, 1.04 mg ZnCl_2_,0.4 mg CoCl_2_·6H_2_O, and 0.25 mg NaMoO_4_·2H_2_O in 100 mL DMDW). The prepared media solution was sterilized at 121 °C for 15 min using an autoclave and, after cooling, it was preserved as a stock solution in the refrigeration (4 °C) temperature. For agar solid media preparation, 5 mL media solution and 75 mg agar were solubilized in a test tube and autoclaved at 121 °C for 15 min for disinfection. The disinfected test tube containing hot media was left on an UV cleaning bench at room temperature in an inclined mode for at least three hours for cooling and solidifying the media into the test tube. Lastly, on the prepared solid media, the microalgae strain (*Pseudokirchneriella subcapitata* NIES-35 collected from National Institute for Environmental Studies, Japan) was applied and allowed to propagation in an illuminated incubator at 13,000 to 14,000 l×, at 22 °C. After 3 days of incubation, cells propagated and were collected as well as washed twice by a centrifugation at 4000 rpm for 10 min with distilled water (DW). Then, thirty milligram (30 mg) microalgae cells were used in 200 μL DW for suspension preparation.

### 2.4. Immobilization of Microalgae

Microalgae cells were immobilized between two filter membranes. Lower filter membrane faced with oxygen sensor array should insulate light in order to protect fluorescence dye from light for photosynthesis. Therefore, we used black MCE filter membrane. Upper filter should pass light in order to accelerate photosynthesis. We employed Omnipore® filter membrane which become half-transparent in wet condition.

The illustration in Figure 2 shows the orderly procedure of microalgae immobilization.

The immobilization procedure was very simple, that is, onto an MCE membrane (diameter 25 mm, pore size 0.45 μm, black type, A045N025A, Advantec, Tokyo, Japan), a space (5 × 5 mm^2^) was created by attaching double adhesive tape, and around 20 μL cell suspension (3 mg cells) was dripped into the space, aiming for an equal cell density in all parts of the space. After that, an Omnipore® membrane (diameter 13 mm, pore size 1.0 μm, white type, 2-305201, Millipore, Burlington, MA, USA) was placed just above the cell layer and attached closely to the double adhesive tape. The cells were trapped between MCE and Omnipore® filter membranes and this is referred to as the biofilm.

### 2.5. Biosensor Chip Fabrication

The above prepared biofilm was then attached closely onto the microarray spots of oxygen sensor chip using a drop of DW, and the edges of biofilm were also fixed onto the planar surface of chip by scotch tape. Above the biofilm, a flexiPERM (11 × 7 × 10 mm^3^, Grainer Bio-One, Frickenhausen, Germany) was also set as a sample holder. Figure 3 shows the detailed illustration of biosensor chip fabrication.

### 2.6. Samples Preparation

We tested several solvents and found 1% DMSO (dimethyl sulfoxide) aqueous solution showed no effect on algae activity. Therefore, 1% DMSO was used for preparation of pesticide solutions. Pesticide solution of 0 ppm means only 1% DMSO was added. Dissolved oxygen (DO) was controlled by using O_2_ and N_2_ gas bubbling and for total deoxygenation of samples, 5% Na_2_SO_3_ was used. Individual stock solution (10,000 mg/L) of each pesticide was prepared and a dilution method was used as for preparing any required lower concentrations.

### 2.7. Instrumentation and Measurement

An inverted fluorescent microscope was used to measure the FI in which dye of microarray oxygen sensor was excited by a light at 450–490 nm wavelength and emission images were collected at above 520 nm wavelength by a high sensitive camera (ORCA-ER C4742-80-12AG, Hamamatsu Photonics, Hamamatsu, Japan). During the experimental period, the room temperature (25 °C) was maintained using an air conditioner (AC) at least 30 min before starting. The oxygen sensibility by our developed microarray oxygen sensor chip was checked by injecting (50 μL of each) different DO having samples through a sample path flow (5 × 7 mm^2^) that was constructed by attaching two 0.2 mm thick SR strips while keeping inside the microarray dye spots of the microarray oxygen sensor and by covering with a cover glass. The average FI of 9 dye spots was measured and the radiometric relationship I_x_/I_30_ (I_x_ means FI intensities of different DO-containing samples, and I_30_, FI of only 30 mg/L DO-containing sample) of samples was plotted.

At the beginning of the pesticide measurement, pesticide-free (only DW) sample (396 μL) was injected into the flexiPERM (sample holder) and covered by a cover glass and subsequently allowed for light/dark (5/5 min) duration. A lamp (with 15,000 l×) was set for supply of lighting. After that, 5 min FI responses due to photosynthetic activity of green algae were monitored where ON–OFF switching (electronic shutter) of the inverted microscope was used to frequently excite dye spots of the biosensor. The procedure can be described as using the ON switch of the electronic shutter, where the dye was excited from the back site and suddenly took an emission image that is called the image at 0 min time. After that, the excitation light was kept off for up to one min but after taking the 0 min image, the upside supply light was kept on. Just a few seconds before 1 min of time, the upside supply light was turned off and, again, the dye was excited and took the emission image that we called image at 1 min time. By following the above method, emission images were taken for up to 10 min. For each emission image, 9 FI measured dye spots were selected, and an average intensity was calculated. The sensor response curves were plotted as FI ratios (I_t_/I_0_) (where I_t_ is intensity at t min and I_0_ is intensity at 0 min) against time. The same procedure was also followed for samples with different concentrations (0 to 100 ppm) of pesticides. Based on the response curves, the rates of FI decrease before and after injection of pesticides samples were evaluated. The inhibition of the oxygen generation rate due to pesticide inhibitors was easily calculated, and their relationship was determined considering the maximum inhibition rate at 5 min time.

## 3. Results and Discussion

### 3.1. Features of Mold, PDMS Stamp, and Printed Dye

Features like size, pitch, and height of spots of patterned mold and PDMS stamp were measured by ImageJ from the electron micrograph. For mold, the average spot size of 97.42 ± 1.00 μm (n = 6), 198.71 ± 1.48 μm pitch (n = 3), and 24.11 ± 1.11 μm (n = 6) spot height were detected. In addition, an average spot size of 98.71 ± 1.13 μm (n = 6), 199.68±1.12 μm pitch (n = 3), and 24.07 ± 1.11 μm (n = 6) spot height were recorded in the case of the PDMS stamp. The prepared patterned mold showed an almost similar size for every spot and it could be used for many times after one successful development. On the other hand, the developed PDMS stamp has good adhesive properties and can easily attach onto the glass substrate. The uniformity of spot heights and top surface of PDMS stamp were clearly noticed, which also helped for quality printing of dye onto the SO_3_-modified glass substrate by the μCP. The measured average size of each spot 97.75 ± 4.50 μm (n = 9) and pitch 205.79 ± 1.38 μm (n = 6) were also documented after printing of dye. It was also observed that the microarray oxygen sensor chip could be used for a long time (several months) by storage in dark conditions and at room temperature (25 °C).

### 3.2. Microarray Oxygen Sensor Chip and Sensibility

The microarray oxygen sensor chip is an optical-based sensor with many advantages, such as easy oxygen permeability into the printed dye, definite measuring point selection, and typical responses. At the excited state, the FI varies due to the concentration of DO, and sensibility in terms of radiometric FI analysis can be easily evaluated. In the high DO sample, the FI is low due to quenching activity between the oxygen and fluorophore molecules, but high in the deoxygenated sample because of the absence of quencher. The experimental results showed that DO from 30 to 0 mg/L has a 1.6-fold sensitive nature, as presented in Figure 4.

### 3.3. Optimum Responses

In the biofilm, microalgae were trapped between two membranes and the cell layer thickness was almost equal to the thickness of double adhesive tape. The membrane (Omnipore®) with large upper pores (5.0 μm) showed easy sample diffusion, i.e., high oxygen diffusion and, thus, the biosensor showed high and smooth responsiveness compared to the small pore (0.45 μm)-based white MCE membrane. In the case of the Omnipore® membrane-based sensor, the reversibility responses are stable but absent for the white MCE membrane-based sensor. Furthermore, initiation of photosynthetic activity or rapid oxygen production depends on the accurate supply and entering of top lights to the microalgae. It is well known that in wet conditions, the used Omnipore® membrane becomes translucent which helps to supply light easily to the microalgae of biofilm. After maximizing the cells densities (3 mg), then for identifying the optimal photosynthesis activity of microalgae by our biosensor chip, two illuminated lights (4000 l× and 15,000 l×) were applied and it was found that 4000 l× was able to initiate photosynthesis reaction poorly in the case of the Omnipore® membrane-based sensor but completely absent for the white MCE membrane-based sensor. Apparently, use of 15,000 l× led to acquiring good responses by photosynthetic activity of microalgae than 4000 l×, and the comparison results are shown in Figure 5A,B.

Furthermore, the recovery responses of Omnipore® membrane-based sensor at dark/light (1/1 or 2/2 min pulses) situations were determined. Figure 6 shows the results of repetitive measurements of O_2_ production under light irradiation (photosynthesis) and O_2_ consumption in the dark (respiration). We found 5% and 10% increasing/decreasing of responses at 1 and 2 min time pulses, respectively. The latter condition was suitable for evaluation of photosynthesis and respiration activities.

### 3.4. Inhibition Measuring Protocol

As phototropic organisms, green microalgae convert light into chemical energy by photosynthesis and generate oxygen as a byproduct. Basically, at photosystem II (PS II), the chlorophyll molecules absorb photons at the start of the electron transport chain, which is the key mechanism of algal photosynthesis reaction. Pesticides are chemicals that usually block the functions of PS II and, as a result, the oxygen generation rate is decreased. The reduced oxygen generation rate could be easily monitored by highly sensitive fluorescence signaling that varies with DO concentrations as well as that the inhibition could be measured by differences in FI, whose rates change under nontoxic (control) and toxic conditions. Generally, the decrease in FI rate is high under nontoxic conditions, i.e., high oxygen production rate, but low under toxic conditions, such as during the reduction of oxygen production. Inhibition of oxygen generation rate could be easily calculated by applying the following Equation (1).
Inhibition of O_2_ generation rate (%) = {(△O_0_ − △O_c_)/△O_0_} × 100(1)
where △O_0_ is the decreasing in FI from 0 to 5 min without pesticide sample and △O_c_ is the decreasing in FI from 0 to 5 min with pesticide in the sample.

### 3.5. Pesticide Sensing and Sensitivity

Figure 7 shows the basic response curves of several diuron concentrations (0–100 mg/L). Standard deviation of FI for 9 spots in the same biosensor was 11%–12%.

The graph clearly describes that in the control without diuron, the decreasing in FI rate or oxygen generation rate is high, but as diuron concentrations increase, there is a reduced rate of FI decrease or oxygen generation because of photosynthetic activity inhibition. Eventually, after 5 min exposure time, no noticeable changes were observed, that is, the decrease in FI rate levelled off and, thus, for sensitivity response identification, 5 min was also selected as the time to evaluate the performance of biosensor in other pesticides (in Supporting Information, a graph of biosensor response to different concentrations of atrazine is also shown in Appendix A). A comparison calibration curves of 6 pesticides are shown in Figure 8. The incubation time for each pesticide was 10 min. At first, the sensor chip was incubated with pesticide solution for 5 min in the dark, then followed by 5 min incubation under light irradiation. Biosensor responses to different pesticides were measured, and the relationship was drawn by plotting the inhibition of oxygen generation rate (%) as a function of individual pesticide concentrations. 

This demonstrates that the minimum concentration of an individual pesticide that initiated inhibition of photosynthetic activity was 1, 10, 10, and 10 ppb for diuron, atrazine, simetryn, and simazine, respectively. However, mefenacet, and pendimethalin did not show any inhibitory activity. Since EC_50_ values reported by Japan Environmental for Agency for diuron, atrazine, simetryn, and simazine are 13, 43, 28, and 42 ppb, respectively, the differences in detection limit should be reasonable.

## 4. Conclusions

A simple photolithographic technique was used to develop a PDMS stamp, which also facilitated good-quality printing of ruthenium complex dye onto a glass slide substrate. The printed microarray spots showed high sensitivity to oxygen when FI was counted at 30 and 0 mg/L DO in samples. Membrane filters completely immobilized the microalgae, and in case of pesticide monitoring, the biosensor depicted good sensitivity in four pesticides samples, like diuron, simetryn, simazine, and atrazine, but was insensitive in mefenacet and pendimethalin samples. The obtained results suggest the highest sensitivity of biosensor in diuron (detection limit: 1 ppb) whereas this was 10 ppb for simetryn, simazine, and atrazine. The detection limit for diuron in conventional biosensors varies from 2.5 to 100 ppb [31,32,33,34]. The present study shows higher sensitivity compared with conventional biosensors. Our developed microalgae biosensor have many advantages like of being simple, low cost, rapid (10 min per sample), requiring only minimal sample volume (200 μL), and also disposable—advantages that are sorely needed for frequent water quality monitoring. In the present study, the biosensor was only exposed to individual pesticides. The next stage is to expose the biosensor to a mixture of pesticides. The present optical oxygen sensor chip was easily prepared using a simple stamp process.

## Figures and Tables

**Figure 1 biosensors-09-00133-f001:**
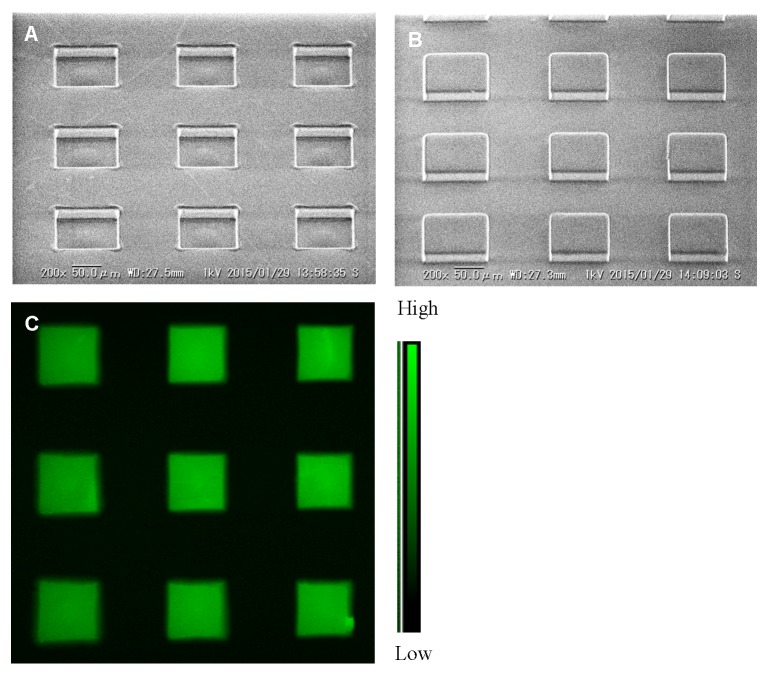
Microarray oxygen sensor chip preparation with the development of stamp mold and polydimethylsiloxane (PDMS) stamp. (**A**,**B**) represent the SEM images of the developed mold and PDMS stamp at 45° angle (subsequently captured by Keyence-7800 microscope), whereas (**C**) indicates printed microarray oxygen sensing dye spots onto SO_3_-modified glass substrate (captured by fluorescence microscope).

**Figure 2 biosensors-09-00133-f002:**
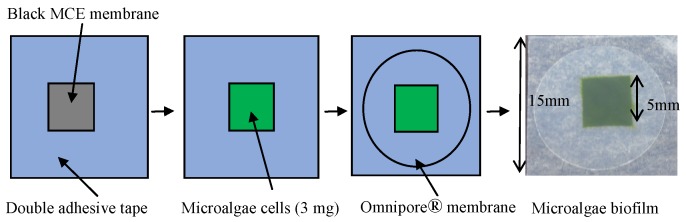
Steps for immobilizing of microalgae within two filter membranes (black mixed cellulose ester (MCE) and Omnipore®).

**Figure 3 biosensors-09-00133-f003:**
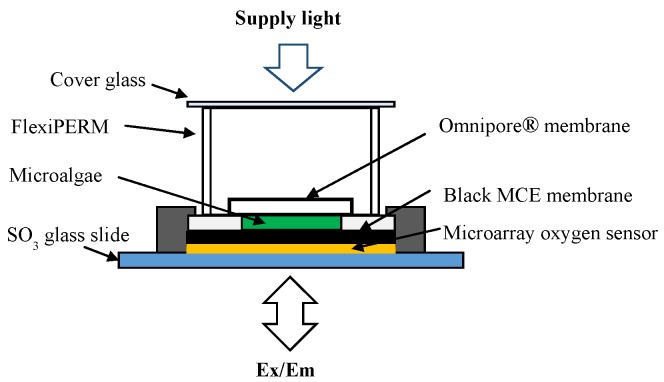
Schematic illustration of the pesticides sensing microalgae biosensor chip.

**Figure 4 biosensors-09-00133-f004:**
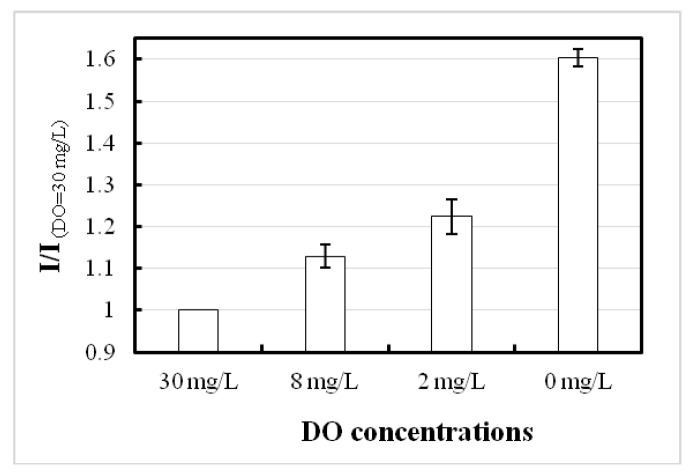
Dissolved oxygen (DO) sensitivity characterization of the microarray oxygen sensor chip.

**Figure 5 biosensors-09-00133-f005:**
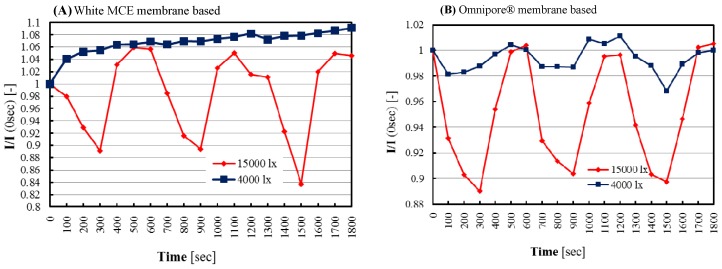
Responses of microalgae biosensors at 4000 and 15,000 l× illuminated conditions, (**A**,**B**) represent microalgae biosensors based on either white MCE or Omnipore® membranes (upper cover), respectively.

**Figure 6 biosensors-09-00133-f006:**
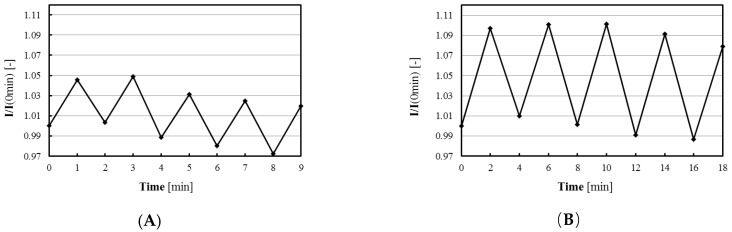
Recovery responses of microalgae biosensor at dark/light pulses. (**A**,**B**) represent 1 and 2 min duration recovery responses, respectively.

**Figure 7 biosensors-09-00133-f007:**
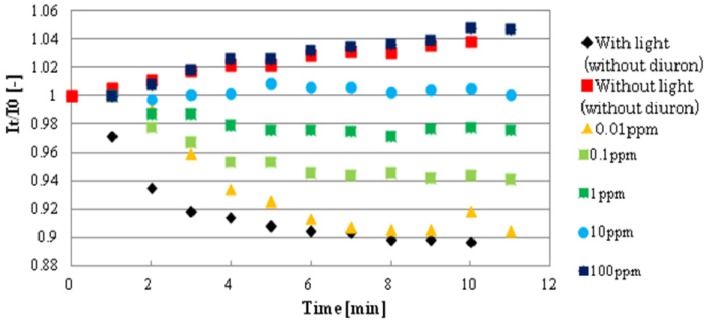
Biosensor response curves in different diuron concentrations (0–100 ppm).

**Figure 8 biosensors-09-00133-f008:**
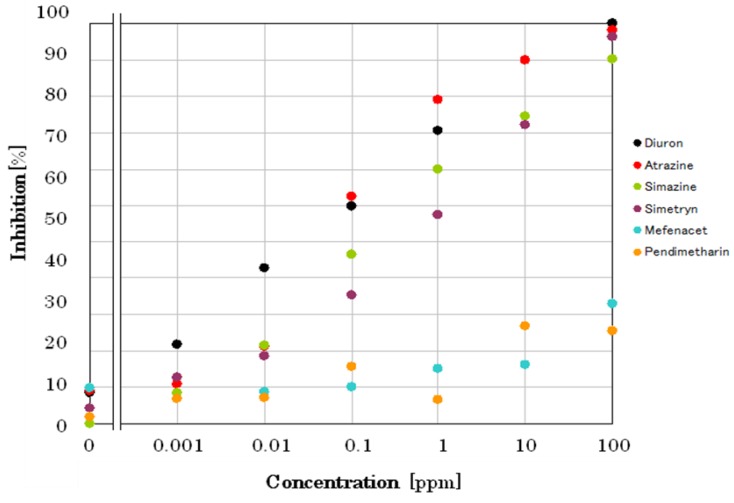
Calibration curves of six pesticides (diuron, simazine, atrazine, simetryn, mefenacet, pendimethalin) versus inhibition of oxygen generation rate (%).

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
