# Peer review of "Development of Microalgae Biosensor Chip by Incorporating Microarray Oxygen Sensor for Pesticides Sensing"

_biosensors, 2019, doi:10.3390/bios9040133_

Round 1

Reviewer 1 Report

The manuscript describes the development and performances of microalgae (Pseudokirchneriella subcapitata) optical biosensor chip for pesticides sensing. The manuscript reports on good sensitivity for four pesticides: diuron, simetryn, simazine, and atrazine; while, insensitive for mefenacet and pendimetharin..

Please refer to the following comments:

In the results and discussion section (page 8, lines 355-366) Figure 7, please add error bars to the graph. Moreover, since the measurements are discrete, you need to omit the lines between the dots. Please add to the supporting information graphs of biosensor response to different concentrations of simazine, atrazine, simetryn, mefenacet and pendimetharin (such as Figure 7 for diuron). In the results and discussion section (page 8, lines 378-393) Figure 8, the results should be presented in bars graph or as discrete dots (no lines between the dots). For each concentration of each pesticide, you should add error bars. Moreover, you need to add control of concentration 0 ppm. Also, please add t-test or ANOVA in order to calculate the statistical significance of the results. What is the limit of detection of the biosensor? How did you calculate the fifty percent inhibition? Can you show it on the graph of Figure 8? Did you try to expose the biosensor to a mixture of pesticides and not only to an individual pesticide? Does the response change?

Author Response

Thank you very much for your appropriate indications and kind suggestions. In the revised version, the manuscript was improved according to your useful comments. Please see the attachment.

Reviewer 2 Report

Review « Development of Microalgae Biosensor Chip by Incorporating Microarray Oxygen Sensor for Pesticides Sensing »

The paper is strongly interesting because it highlights the use of a novel technique as powerful tool for pesticides sensing. The different steps of the construction of the biosensor seem clearly stated. However, several points need to be developed, corrected and clarified.

As it is mentioned in the introduction, there is plenty of papers about biosensors for toxicity sensing, and I guess particularly using diuron, simazine, and atrazine at least… Could a few words be added to have a state of the art about the sensibility of the other biosensors to these compounds and thus, the motivation of designing a new biosensor? In the material and methods part, chemicals are listed by a strange way (all together). Maybe it should be better organized by kind of products? If the authors want to be exhaustive, the origin of only two pesticides used here are mentioned (references for diuron, simetryn, mefenacet, pendimetharin… are missing). In results and discussion part, the main difficulty for me is to replace the use of this biosensor in the context of all other existing biosensors dedicated to the sensing the pesticides studied here. In what this biosensor will bring something more robust or precise or performing than other already published biosensors? It is obviously due to the rapidity of the analysis, and it could more emphasized by a rapid comparison with the techniques classically employed in the text and perhaps also as an illustration (maybe in a Table format)? The same kind of question concerning the sensitivity of this biosensor for diuron (13 ppb) compared to what is found in the bibliography… Can the authors compare this value to the ones traditionally obtained by other biosensors? This could be added in the same Table (see my previous question). In none of the graphs there are standard deviations. Could you please add these indications? It will enhance the credibility of the results. I am not sure to understand the utility of Fig. 6. What does it imply exactly? Do you advise to use 2/2 min pulse? I don’t really understand how the microalgae produce oxygen in the absence of light and in presence of diuron (red curve, Fig. 7)? A non-toxic case? At which concentration is the diuron in this case? Pesticides were diluted with a solvent (dimethyl sulfoxide-DMSO). Thus, it appears necessary to perform controls with this compound in order to check that it has no negative impact on the microalgae photosynthesis and oxygen emission, and to be sure that the observed effects are only due to the pesticide alone. Clearly, why the authors did not add a curve of test with DMSO and without pesticide in Fig. 8? It is not very clear from the material and methods section that the “pesticide free” sample used at time 0 was a blank for all along the experiment? The sentence L.352 to L.354 is stated for Fig. 8. It should be mentioned inside brackets, and the sentence should appear later in the text. The biosensor is presented as rapid test (15 min per sample, L.403). Therefore, I was wondering why the time-course experiment in Fig. 7 does not go until 15 min. In relation to the previous question, it is not really clear for me, at which incubation time the data presented in the curves from Fig. 8 were obtained. Is it 5 min (like I suppose form the text in material and methods, L. 255), 10 or 15 min ? It is definitively not clear… I strongly suggest the authors to read carefully the manuscript and perform extensive editing of English. There is too numerous English mistakes, first of all in the abstract and the introduction. All together, these grammatical mistakes strongly downgrade the paper. As examples, the first sentence of the abstract is too long, and I suggest to divide it in two parts (there are other examples all along the introduction).

Some mistakes and corrections:

L.19 “… the biosensor showed…”

L.20 “… but was insensitive for…”

L.30 “…758 metric tons were used…”

L.40 “… 4 million people suffering…”

L.44 “… are found now frequently and suspected to have a direct link…”

L.71 “Immobilization” instead of “immobilizing”

L.298 “… that helps to supply light easily…”

L.340 “… as a result the oxygen generation rate is decreasing.”

L.368 “diuron

etc….

Author Response

Thank you very much for your appropriate indications and kind suggestions. In the revised version, the manuscript was improved according to your useful comments. Please see the attachments.

Round 2

Reviewer 1 Report

Thank you for the clear answers.

I believe that the manuscript is ready to be published in its present form.

Author Response

Thank you very much for your excellent reviewing.

With best regards.

Reviewer 2 Report

Dear authors,

Thank you for your answer and corrections for this work of high quality. Your adds enhance the understanding of both the context and the methodology. This is now fine for me to accept your manuscript for publication after these last minimal corrections.

L. 20 space between "1" and "ppm"

L. 386 "detection"

L. 397 space between "100" and "ppm"

L. 401 "to individual pesticide only"

L. 401 "We need to expose"

In addition, I am not sure the last sequence is useful as I can't see how the chip "could be measured thanks to a mobile phone"

Best regards

Author Response

Thank you very much for your excellent reviewing and suggesting exact our mistakes.

All the points you suggested us were corrected in the revised manuscript.

The last sequence concerning a mobile phone was deleted in the revised manuscript.

With best regards.